# The Thermo-Oxidative Behavior of Cotton Coated with an Intumescent Flame Retardant Glycine-Derived Polyamidoamine: A Multi-Technique Study

**DOI:** 10.3390/polym13244382

**Published:** 2021-12-14

**Authors:** Claudia Forte, Jenny Alongi, Alessandro Beduini, Silvia Borsacchi, Lucia Calucci, Federico Carosio, Paolo Ferruti, Elisabetta Ranucci

**Affiliations:** 1Institute of Chemistry of OrganoMetallic Compounds (ICCOM), National Research Council, Via G. Moruzzi 1, 56124 Pisa, Italy; silvia.borsacchi@pi.iccom.cnr.it (S.B.); lucia.calucci@pi.iccom.cnr.it (L.C.); 2Dipartimento di Chimica, Università degli Studi di Milano, Via C. Golgi 19, 20133 Milano, Italy; jenny.alongi@unimi.it (J.A.); alessandro.beduini@unimi.it (A.B.); paolo.ferruti@unimi.it (P.F.); 3Dipartimento di Scienza Applicata e Tecnologia, Politecnico di Torino, Alessandria Campus, Viale T. Michel, 15121 Alessandria, Italy; federico.carosio@polito.it

**Keywords:** linear polyamidoamines, cotton thermal oxidative decomposition, char, ^13^C MAS NMR, Raman, XPS

## Abstract

Linear polyamidoamines (PAAs) derived from the polyaddition of natural α-amino acids and *N*,*N*′-methylene bis(acrylamide) are intumescent flame retardants for cotton. Among them, the glycine-derived M-GLY extinguished the flame in horizontal flame spread tests at 4% by weight add-on. This paper reports on an extensive study aimed at understanding the molecular-level transformations of M-GLY-treated cotton upon heating in air at 300 °C, 350 °C and 420 °C. Thermogravimetric analysis (TGA) identified different thermal-oxidative decomposition stages and, coupled to Fourier transform infrared spectroscopy, allowed the volatile species released upon heating to be determined, revealing differences in the decomposition pattern of treated and untreated cotton. XPS analysis of the char residues of M-GLY-treated cotton revealed the formation of aromatic nanographitic char at lower temperature with respect to untreated cotton. Raman spectroscopy of the char residues provided indications on the degree of graphitization of treated and untreated cotton at the three reference temperatures. Solid state ^13^C nuclear magnetic resonance spectroscopy (NMR) provided information on the char structure as a function of the treatment temperature, clearly indicating that M-GLY favors the carbonization of cotton with the formation of more highly condensed aromatic structures.

## 1. Introduction

Intumescent flame retardants (IFRs) represent a promising family of environment friendly flame retardants owing to their high efficiency usually associated with low smoke emission and toxicity [1,2,3,4]. Presently, most IFRs contain three major components: an acid source, a carbonizing source, and a blowing agent. The acid source can be a strong acid, such as sulfuric acid and phosphoric acid; the carbonizing source includes different oligosaccharides and char-forming polymers, whereas the blowing agent is normally a nitrogen-containing compound, including urea, melamine, and urea-formaldehyde resin. At high temperatures, the acid source decomposes and generates inorganic strong acids, which promote the dehydration of the carbonizing agent to produce the carbonaceous layer. Meanwhile, the blowing agent degrades and releases non-flammable gases, which expand the carbonaceous layer forming a swollen multicellular layer. This layer can insulate the underlying polymer and prevent or, at least limit, the exchange of heat, oxygen and combustible volatiles, thus reducing the possibility of flame propagation.

In the literature, there are various studies that propose single polymers, belonging to the polyamidoamines (PAAs) family, carrying in their main chain groups that confer all the above properties as all-in-one-solutions [5]. PAAs are multifunctional polymers obtained by the aza-Michael polyaddition of *prim*- or bis-*sec*-amines with bis-acrylamides [5,6]. The reaction is usually carried out in water or protic solvents at room temperature, pH 8–10, and in the absence of added catalysts. Under these conditions, the reaction is highly selective since, in addition to the *prim*- or *sec*-amines, only the thiol and phosphine groups can give 1,4-conjugate addition with acrylamides. Many PAAs, besides showing remarkable potential as IFRs for cotton fabrics [7,8,9,10,11], are biocompatible and, therefore, have been extensively studied for biotechnological applications [6,12,13]. In particular, several PAAs derived from natural α-amino acids were found to be non-flammable by applying a butane flame: intumescence occurred on their surface, which swelled and blackened, while the interior remained apparently unaltered.

It was found that PAAs extinguished the flame in horizontal flame spread tests (HFSTs) at add-ons ranging from 4% to 19% based on the structure of the amine sub-units: 4% for glycine-derived PAA [7], 5% for glutamine- and glutamic acid-derived PAAs [11], and 19% for PAAs carrying guanidine pendants [7]. In vertical flame spread tests (VFSTs), α-amino acid-derived PAAs normally failed to extinguish the flame but left a substantial residue. Among α-amino acid-derived PAAs, the glycine-deriving one, M-GLY, was one of the most efficient in imparting self-extinguishment to cotton in HFSTs and, in oxygen-consumption cone calorimetry tests, significantly increased the resistance to an irradiative heat flux of 35 kWm^−2^ [7]. Moreover, the glycine/*L*-cystine copolymers, coded M-G-C, also effectively extinguished the flame both in HFSTs and VFSTs [10].

In previous works, it was found that M-GLY extensively intumesced when heated in air at 350–500 °C [7,11]. In HFSTs, performed on cotton treated with M-GLY with 4% add-on, the samples initially ignited, but then extinguished, leaving approximately 82% residual mass fraction, most of which intact, while the residual portion consisted of a carbonaceous residue that had maintained the original texture of the fabric. SEM analysis showed that the fibers present in the burned area were coated with intumescence. It is worthy of note that, in the same temperature range, cotton underwent the main phase of thermal oxidative degradation, suggesting that the FR properties of M-GLY were due to the occurrence of intumescence that hindered flame propagation.

The objective of this work was to understand the molecular-level transformations of M-GLY-treated cotton upon heating in air at 300 °C, 350 °C and 420 °C and comparing them with those of untreated cotton. To this purpose, several techniques were used to compare the effect of heating in air on pristine cotton (COT) and cotton impregnated with M-GLY (COT/M-GLY). In particular, thermogravimetric analysis (TGA) was applied to highlight the main stages of the thermo-oxidative degradation of cotton, TGA coupled to Fourier transform infrared spectroscopy (TG-IR) allowed volatiles produced during heating to be identified, while Raman, X-ray photoelectron (XPS), and ^13^C solid state nuclear magnetic resonance (SSNMR) spectroscopies provided details on the structural changes undergone by COT and COT/M-GLY upon heating in air.

## 2. Materials and Methods

### 2.1. Materials

Glycine (GLY, 98%), *N*,*N*′-methylene bis(acrylamide) (MBA, 99%), lithium hydroxide monohydrate (98%), and D_2_O (99.9%) were purchased from Sigma-Aldrich (Milano, Italy) and used as received. Woven and scoured cotton fabric having an area density of 200 g m^−2^ was purchased from Fratelli Ballesio S.r.l. (Torino, Italy).

### 2.2. Synthesis of M-GLY

M-GLY was synthesized as already reported [7]. In brief, MBA (15.40 g; 0.10 mmol), GLY (7.50 g; 0.10 mmol) and lithium hydroxide monohydrate (4.20 g; 0.10 mmol) were dissolved in water (35 mL). The reaction mixture was heated at 40–45 °C until complete dissolution of MBA and then gently magnetically stirred for 5 days at room temperature in the dark under nitrogen. It was then diluted to 300 mL with water, the pH adjusted to 4.5 with 37% hydrochloric acid and ultra-filtered through a membrane with nominal molecular weight cutoff 5000 Da. The product was finally retrieved by freeze-drying the retained portion. Yield: 19.5 g (85.0%). The amount of LiOH·H_2_O was not included in the yield calculation since it was lost during the purification and isolation procedures.

### 2.3. Impregnation of Cotton Fabrics with M-GLY

Stripes of cotton fabrics of the appropriate dimensions were dried for 2 min at 100 °C and weighed. A 5 wt.-% PAA aqueous solution was uniformly drop-wise distributed on the specimens. After deposition, samples were dried 5 min at 100 °C. The total dry solid add-on (wt.-%) was determined by weighing each sample before (*W_i_*) and after impregnating with the PAA solution and drying (*W_f_)*, using an analytical balance (±10^−4^ g accuracy). The add-on was calculated according to Equation (1):(1)$$Add-on=\frac{W_{f}-W_{i}}{W_{i}}\times 100$$

The final add-on was 4% for cotton treated with M-GLY. The sample was coded COT/M-GLY.

### 2.4. Sample Preparation for Microscopic and Spectroscopic Analyses

M-GLY (500 mg), untreated and M-GLY-treated cotton fabrics (50 mm × 100 mm each) were placed in porcelain crucibles and heated in air in a Nabertherm B180 oven (Lilienthal, Germany) with a heating rate of 10 °C min^−1^ up to 300 °C, 350 °C and 420 °C, respectively, and then held for 2 min at this temperature before being cooled to room temperature.

### 2.5. Size Exclusion Chromatography

Size exclusion chromatography (SEC) traces were obtained for all polymers with Toso-HaasTSK-gel G4000 PW and TSK-gel G3000 PW columns connected in series, using a Waters model 515 HPLC pump (Milano, Italy) equipped with a Knauer autosampler 3800 (Knauer, Bologna, Italy), a light scattering (670 nm), a viscometer Viscotek 270 dual detector (Malvern, Roma, Italy), and a refractive index detector (Model 2410, Waters, Milano, Italy). The mobile phase was a 0.1 M Tris buffer (pH 8.00 ± 0.05) solution with 0.2 M sodium chloride (sample concentration: 20 mg mL^−1^; flow rate: 1 mL min^−1^; injection volume: 20 μL; loop size: 20 μL; column dimensions: 300 mm × 7.5 mm). The instrument optical constants were determined using PEO 24 kDa as a narrow standard. Before analysis, each sample was filtered through a 0.2 μm WhatmanTM syringe filter (Maidstone, UK).

### 2.6. Thermal Properties

Thermogravimetric analyses (TGA) were obtained from 35 to 800 °C in air using a Mettler-Toledo TGA/DSC 2 Star System instrument (Milano, Italy). Cotton samples (5–10 mg) were placed in alumina crucibles and heated with a 10 °C min^−1^ heating rate under a 50 mL min^−1^ air flow. In addition, in order to monitor the volatile species released upon heating, thermogravimetric analyses coupled with infrared spectroscopy (TG-IR) were carried out using a Perkin Elmer TGA 4000 microbalance (Milano, Italy) interfaced to a Perkin-Elmer Frontier FT-IR/FIR spectrophotometer (Milano, Italy). Cotton samples (5 mg) were placed in alumina crucibles and heated with a 10 °C min^−1^ heating rate under a 20 mL min^−1^ air flow. Spectra were recorded at intervals of 2.8 s in the 4000–500 cm^−1^ spectral range using a resolution of 4 cm^−1^.

### 2.7. Scanning Electron Microscopy

The surface morphology of untreated and impregnated cotton and combustion residues was studied using a LEO-1450VP scanning electron microscope (SEM) (5 kV beam voltage, 15 mm working distance). M-GLY, COT and COT/M-GLY residues (5 mm × 5 mm) were fixed to conductive adhesive tapes and gold-metallized.

### 2.8. Raman Spectroscopy

Raman spectra were performed on M-GLY, COT and COT/M-GLY residues by using a Renishaw InViaTM Raman (Gloucestershire, UK) equipped with an argon laser (514 nm wavelength, 50 mW power, 3 scans) coupled with a Leica (Wetzlar, Germania) DM 2500 optical microscope.

### 2.9. X-ray Photoelectron Spectroscopy

M-GLY, COT and COT/M-GLY residues were analyzed using an X-ray photoelectron spectrometer (XPS) equipped with an Al K radiation monochromatic source (1486.6 eV) and manufactured by Surface Science Instruments—Biolin Scientific UK (Manchester, UK).

### 2.10. Solid State Nuclear Magnetic Resonance

Solid state NMR (SSNMR) experiments were carried out on a Bruker Avance Neo spectrometer working at 500.13 MHz for ^1^H and at 125.77 MHz for ^13^C, equipped with a 2.5 mm triple-channel CP-MAS probe head. ^1^H-^13^C cross-polarization (CP) NMR experiments with magic angle spinning (MAS) were recorded under high power proton decoupling conditions, with a 90° pulse length of 2 μs, a contact time of 2 ms, a MAS frequency of 15 kHz, and a recycle delay of 2 s. For the spectra, 14,800 to 32,000 scans were accumulated depending on the sample.

## 3. Results and Discussion

### 3.1. Synthesis of M-GLY

The PAA coded as M-GLY has been obtained by the aza-Michael polyaddition of *N*,*N*′-methylene bis(acrylamide) with glycine, carried out in water at pH 9–10, at 40–50% concentration on a weight basis and at room temperature (Figure 1), as previously reported [7]. The structure of M-GLY was confirmed by ^1^H-NMR and FT-IR spectroscopies (Appendix A). The number average molecular weight, ${\bar{M}}_{n}$, as determined by size exclusion chromatography (SEC) was 7500, with polydispersity index 1.4. M-GLY is a water soluble amphoteric PAA with isoelectric point 4.5.

### 3.2. Thermogravimetric Analysis of M-GLY-Treated Cotton

As previously shown [7], comparing the TGA curve of 4% add-on M-GLY-treated cotton (COT/M-GLY) with that of untreated cotton (COT) (Figure 1a), it is apparent that the onset decomposition temperature at 10% weight loss, T_onset10%_, of COT/M-GLY (286 °C), as well as the temperature at maximum weight loss rate, T_max_ (320 °C), were lower than those of COT (T_onset10%_ 315 °C, T_max_ 349 °C) (Table 1 and Figure 1b). However, the residual mass fraction of COT/M-GLY at 400 °C was significantly higher than that of COT (30% versus 22%), suggesting that M-GLY sensitized cotton with regards to triggering thermal decomposition, making it at the same time less subject to oxidative degradation at higher temperatures. Based on the TG curves of Figure 1a, three temperature values were identified, namely 300 °C, 350 °C and 420 °C, corresponding to different phases of the thermo-oxidative decomposition of cotton, for both COT/M-GLY and COT. The first temperature value (300 °C) was just before the major decomposition step; the second (350 °C) roughly corresponded to the inflection of both TG curves. The third temperature value (420 °C) corresponded to the second intersection of the two TG curves, after the major decomposition step. It was considered particularly interesting to analyze the residues obtained after heating to these temperatures by means of different techniques.

### 3.3. TG-IR Analysis

The TG-IR analysis of untreated and 4% add-on M-GLY treated cotton (Figure 2) allowed to compare the nature and the relative proportion of the gases released from both samples during a heating ramp from 50 °C to 800 °C.

The thermal-oxidative degradation mechanism of cotton has already been the subject of many investigations [14]. Briefly, three main stages have been identified. The first stage, which occurs from 300 °C to 400 °C, involves two competing pathways that produce volatile substances and aliphatic oxygenated char. In the second phase, which occurs from 400 °C to 800 °C, part of the aliphatic char evolves towards aromatic structures. The chars produced in both stages undergo partial oxidation releasing CO and CO_2_. Finally, above 800 °C the chars formed in the previous stages are further oxidized to CO_2_ and, to a lesser extent, to CO. Volatiles include small amounts of water, resulting from dehydration reactions, aliphatic hydrocarbons, as well as ketones and aldehydes produced by the thermal decomposition of levoglucosan, produced by the decomposition of cellulose. Aliphatic hydrocarbons and carbonyl compounds obviously provide the combustible volatiles feeding the flame. In the present work, it was found that all the above volatiles are released in the thermal-oxidative degradation of both COT and COT/M-GLY samples (Figure 2a,b), but with qualifications. As evidenced in Figure 2c,e, it should be noticed that a significant difference existed between the relative proportion of CO_2_ and carbonyl compounds evolved: the amount of CO_2_ released from M-GLY-treated cotton was higher than that released from untreated cotton, whereas the amount of carbonyl compounds was lower (Figure 2c,e, respectively).

It should be observed that nitrogen-containing volatiles were not detected in the gaseous phase. This was probably due to the very low amount of M-GLY on the cotton surface. It is evident that M-GLY, although present in small quantities, exerted a significant influence on the decomposition process of cotton. In line with what generally reported for flame retardants, M-GLY favors the production of char from levoglucosan and reduces its conversion into aldehydes and ketones [15]. In any case, no evidence has been collected concerning the influence of the M-GLY nitrogen on the degradation mechanism of cotton.

### 3.4. Morphological Analyses of Samples Thermally Treated in Air

Figure 3 shows the aspect of M-GLY, 4% add-on COT/M-GLY and COT heated at 420 °C in air. As previously shown [7], M-GLY undergoes extensive intumescence, giving rise to a voluminous spongy char. At first glance, the residues of cotton and cotton treated with M-GLY appeared macroscopically similar, but SEM analysis revealed significant differences.

The results of the SEM observation of the surface morphology of COT and 4% add-on COT/M-GLY heated at 300 °C, 350 °C and 420 °C are shown in Figure 4. At 300 °C, the texture of the fabrics was substantially maintained and the fibers of both COT and COT/M-GLY preserved the original spiraling of the natural cellulose fibrils. The surface of COT/M-GLY was generally flat and fairly smooth, not significantly different from that of COT. At 350 °C, the texture began to alter, and the fiber distribution became sparser. Meanwhile, the surface of the COT/M-GLY fibers showed small irregularities and protuberances, attributable to M-GLY intumescence. At 420 °C the texture of the COT strips collapsed sharply, and the fibers lost their regular spiraling. In contrast, COT/M-GLY mostly retained both the texture and spiral of the fibers and, furthermore, the surface of the fibers revealed an increase in surface irregularities. As for M-GLY (Figure 5), at 350 ° C a honeycomb structure due to the partial volatilization of the decomposition products was clearly visible. At 420 °C, this structure collapsed, and further cavities appeared due to the release of volatile species.

### 3.5. Spectroscopic Characterization of Chars

To study the influence of M-GLY on the molecular transformations undergone by cotton subject to thermal-oxidative treatment, untreated and 4% add-on M-GLY-treated cotton fabrics were heated in air at the reference temperatures identified by TGA analysis, i.e., 300 °C, 350 °C and 420 °C (see above). All char residues were characterized by Raman, XPS and ^13^C SSNMR spectroscopies. However, as already highlighted by the morphological characterization and TG analyses, the char content of untreated cotton was lower than that of M-GLY-treated cotton. Figure 6 shows the XPS spectra with energy ranging from 200 to 1200 eV of the residues left after heating COT (Figure 6a) and COT/M-GLY (Figure 6b) up to the three reference temperatures considered. Herein, O and C elements were detected as the most abundant by the probe. Figure 6c,d show the expansions of the C 1s band in the 276 and 296 eV regions for COT and COT/M-GLY, respectively. At first glance, it is evident that the spectrum of COT/M-GLY heated at 300 °C resembles that of COT heated at 350 °C, while the spectrum of COT/M-GLY heated at 350 °C resembles that of COT heated at 420 °C. This mutual displacement of the two sets of curves along the temperature axis means that the effect of M-GLY was to anticipate at lower temperatures the structural changes to which cotton is subjected during the thermal-oxidative process. The C1s region of both COT and COT/M-GLY was dominated by the primary signals due to the aromatic C-H and C-C bonds, whose relative contribution increased with increasing the treatment temperature. More in detail, three regions of interest were identified in these spectra (Figure 6c,d): 282–285 eV, 286–288 eV and 288–289 eV. The first region, between 282 and 285 eV, is consistent with the presence of three bands relative to C-C bonds, that is, 283.4–284.0 eV (low), 284.2–284.6 eV (primary) and 284.3–285.4 eV (high) [16]. Particularly the signal at 284.3–285.4 eV, relative to C-C and C-H bonds, is indicative of the aromatic character of the residues. As already observed, the spectral intensity of this signal increased with the treatment temperature. Defects due to small rings, C5 or less, within larger aromatic systems were observed for COT/M-GLY heated at 420 °C, as identified by the shoulder at around 282.5 eV (pointed by a black arrow in Figure 6d) [16]. In the second region, between 286 and 288 eV, the contribution of C-O (285.9–286.6 eV) and C=O bonds (286.7–287.5 eV) is suggested particularly for COT and COT/M-GLY heated at 350 °C. In this region, a reduction in the height of the band was observed with increasing temperature, particularly for cotton, ascribed to the progress of dehydration and oxidation reactions. In the third region, from 288 to 289 eV, signals ascribed to COO groups emerged for cotton above 400°C, while they were already revealed for COT/M-GLY at 350 °C and increased shifting to 420 °C, in line with the progress of the oxidation reaction. Meanwhile the hydroxyl/ether (285.9–286.6 eV) and carbonyl (286.7–287.5 eV) contributions decreased with increasing temperature, consistent with the progress of the dehydration and oxidation reactions.

The Raman spectra of COT and COT/M-GLY are shown in Figure 7a,b. Two peaks can be identified, as it is normally the case for carbon materials: the G peak at 1590 cm^−1^, the so-called graphitic peak, and the D peak at 1360 cm^−1^, which is ascribed to defects. The former peak is ascribed to the bond stretching of all pairs of C sp^2^ atoms in rings and the latter to the chains and breathing modes of C sp^2^ atoms in rings. These peaks are relatively broad, and this is indicative of a distribution of other underlying peaks; in fact, in the recent years there has been a discussion on how to deconvolve these peaks and different approaches have been proposed [17]. In both COT and COT/M-GLY, the position of the G peak and the ratio of the intensities I(D)/I(G) > 0.5 are indicative of the formation of nanographitic structures. Comparing the two histograms that report the area ratio for D and G peaks in Figure 7c,d, it is possible to observe that the amount of D and G areas are comparable at 300 °C and 350 °C. Conversely, at 420 °C in COT/M-GLY the large D bandwidth, together with the small shift observed by increasing temperature, suggests the presence of condensed aromatic structures of different sizes (Figure 7b), whose formation is obviously induced by M-GLY [17,18].

^13^C SSNMR spectra were recorded on residues of COT and COT/M-GLY after heating in air at 300 °C, 350 °C, and 420 °C using the ^1^H-^13^C cross-polarization (CP) technique. In the CP experiment, magnetization is transferred from the most abundant proton nuclei to dipolarly coupled ^13^C nuclei; the observed CP-enhanced ^13^C magnetization results from the competing effects of ^1^H-^13^C polarization transfer and the proton longitudinal relaxation in the rotating frame. As a consequence, quantitative information cannot be extracted directly from a single spectrum recorded using a given set of experimental conditions. Nevertheless, CP spectra acquired with a suitable contact time duration (2 ms in our case) can be safely used to investigate changes in the relative amounts of molecular substructures in chars. As shown in Figure 8, a comparison of the spectra recorded on COT and COT/M-GLY after the thermal treatments clearly indicated that the transformation of cotton by thermo-oxidative processes occurs at lower temperature for COT/M-GLY than for COT, in agreement with the above reported analyses, as well as with results reported in the literature for cotton treated with different flame retardants [19,20]. In fact, the spectrum of COT heated at 300 °C showed quite strong and sharp signals between 60 and 110 ppm arising from cellulose [19,21], the main component of cotton, together with broad peaks from aliphatic (0–60 ppm), aromatic and furanic (110–160 ppm), carboxyl (~167 ppm), and carbonyl (~198 and 205 ppm) groups in materials arising from the transformation of cellulose [22,23,24,25,26,27,28,29,30,31]. Indeed, upon heating in air, cellulose undergoes decomposition of the glycosidic units, dehydration, decarboxylation, and condensation polymerization reactions. These reactions, which occur contemporaneously following different paths, are known to result in alkyl, furanic and aromatic moieties bearing ketone, aldehyde, carboxylic acid and ester functionalities, depending on the treatment temperature. In particular, when heating is carried out at higher temperatures, the alkyl, furanic, carbonyl and carboxyl groups progressively disappear, a progressive loss of hydrogen and oxygen and a related enrichment in carbon is observed, accompanied by an increase in aromatic groups and eventually by the formation of condensed aromatic clusters [22,23,24,28,29,30]. In fact, after heating at 350 °C, no more signals of cellulose were observed in the spectrum of COT, signals from aliphatic groups were very weak, and those from carboxyl and carbonyl groups showed reduced intensity. On the other hand, signals arising from aromatic structures dominated the spectrum; quite sharp peaks were observed at ~129 ppm, ascribable to unsubstituted and H-substituted aromatic carbons, and at 158 ppm, due to aromatic carbons bonded to hydroxyl groups. The signals of carbons ortho and para to oxygen-substituted carbons (120 ppm) underlay the major peak at 129 ppm. A similar spectrum was observed for COT treated at 420 °C. In the case of COT/M-GLY, heating at 300 °C already resulted in the practically complete transformation of cellulose, with only extremely weak peaks found in the ^13^C spectrum. At variance, signals from aliphatic, aromatic, carbonyl and carboxyl groups were clearly observed. The ^13^C SSNMR spectrum recorded on COT/M-GLY treated at 350 °C closely resembled that of COT heated at the same temperature, while COT/M-GLY treated at 420 °C showed signals practically only from aromatic and carboxyl groups.

In order to quantify the relative proportions of carbons in the different functional groups, the corresponding signals were integrated in the relevant spectral regions. The results, shown in Figure 9, confirmed the qualitative description of the spectra: cellulose is still abundant in COT, but almost completely transformed in COT/M-GLY, after heating in air at 350 °C; aromatic groups dominate the spectra of COT heated at 350 °C and 420 °C and those of COT/M-GLY treated at all temperatures, with a smaller proportion of carbonyl and carboxyl groups and weak signals from aliphatic structures, which progressively disappear on increasing the treatment temperature. It should be noted that the aromatic signals exhibited a much lower intensity for COT/M-GLY heated at 420 °C than for the other samples (Figure 8 and Figure 9). Considering that the ^13^C SSNMR spectra were acquired using the CP technique, where ^13^C magnetization is built by transfer from the most abundant ^1^H nuclei, this feature must be ascribed to a depletion of hydrogen atoms bonded or close to carbons in this char, due to a higher condensation of the aromatic structures, and/or to an increase of the proton relaxation rate in the rotating frame caused by the presence of organic free radicals produces by the thermo-oxidative processes [23,24,28,30]. Dipolar dephasing experiments [29,32], in which the trend of the intensity of the aromatic carbon signals is measured as a function of the dephasing time with respect to a standard (see Appendix A for details), clearly indicated the presence of a larger fraction of hydrogens located farther from carbons, and, therefore, of more highly condensed aromatic structures in COT/M-GLY than in COT after heating at 420 °C.

## 4. Conclusions

It has been recently found that the glycine-derived PAA, M-GLY, has a significant flame retardant activity for cotton, being able to extinguish flame in horizontal flame propagation tests with modest add-on. The final residue was largely unburnt. The burnt portion of residue maintained the original texture and, moreover, the fibers showed a high production of microbubbles, as demonstrated by the SEM analysis, revealing an extensive intumescence. The results of oxygen-consumption cone calorimetry tests also revealed a quantitative reduction of the carbon monoxide and dioxide production, as well as of the peak of heat release rate [7]. Interestingly, the onset decomposition temperature of cotton was lower in the presence of M-GLY, that is, M-GLY sensitized cotton to thermal decomposition. On the other hand, M-GLY did not burn by flame impingement, but intumesced superficially. Moreover, the maximum intumescent char production took place at approximately the maximum decomposition temperature of cotton and this phenomenon has obviously a bear in affecting the structural transformations at the molecular level undergone by cotton during thermal-oxidative treatment.

This paper reports on the results of an extensive characterization study of char produced in air from heating untreated and M-GLY treated cotton at three reference temperatures, namely 300 °C, 350 °C and 420 °C, corresponding to different thermo-oxidative decomposition steps, to interpret the aforementioned phenomena at the molecular level. In parallel, the thermal decomposition mechanism was investigated by TG-IR, demonstrating that the M-GLY/cotton interaction influences the nature and amounts of volatiles released by cotton on heating in air.

The char produced from untreated and M-GLY-treated cotton was characterized by XPS, Raman and ^13^C SSNMR spectroscopies. It should be observed that the amount of char produced by M-GLY-treated cotton at the highest investigated temperature, 420 °C, was remarkably higher than that of untreated cotton. However, at this temperature, the texture and fiber morphology of the treated cotton were much better preserved than in the case of untreated cotton, as revealed by SEM.

XPS analysis revealed the formation of an aromatic, possibly nanographitic char, at lower temperature for M-GLY-treated cotton. Raman spectra gave clear indication of graphitization of both treated and untreated cotton between 300 °C and 420 °C. In particular, the large bandwidth of the D band suggested the presence of condensed aromatic structures of different size in M-GLY treated cotton. ^13^C SSNMR gave detailed information on the structures formed by cotton cellulose by the thermo-oxidative processes and demonstrated that M-GLY-treated cotton heated at 420 °C had larger condensed aromatic structures with respect to cotton that underwent the same thermal treatment.

## Data Availability

Not applicable.

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
