# Peer review of "The Thermo-Oxidative Behavior of Cotton Coated with an Intumescent Flame Retardant Glycine-Derived Polyamidoamine: A Multi-Technique Study"

_polymers, 2021, doi:10.3390/polym13244382_

Round 1

Reviewer 1 Report

In this paper fire behavior of pristine cotton and M-GLY treated Cotton are investigated. The improvement of fire retardance in treated cotton is based on intumescence. Mechanisms of carbonization are attentively explored by using a set of different analysis. The paper is clearly written and I think that it can be published with minor corrections, as below indicated-.

line 257 and following "As for M-GLY (Figure 5), the residues at 350 °C clearly showed the spherical cavities  due to the volatilization of part of the decomposition products. At 420 °C this phenomenon was more evident, and the dimensions of the cavities increased considerably". However,  the morphology are greatly different at 350 and 420 °C and in fig 5 I cannot see increasing of their dimensions. Can you highligth  on the  figure where are the cavities which size is increasing?

lines 353 354 please delete the sentence in bold

line 408 409. Please cite the reference because thr cone calorimeter results are not shown in the present paper

Author Response

Reviewer 1

Comments and Suggestions for Authors

In this paper fire behavior of pristine cotton and M-GLY treated Cotton are investigated. The improvement of fire retardance in treated cotton is based on intumescence. Mechanisms of carbonization are attentively explored by using a set of different analysis. The paper is clearly written, and I think that it can be published with minor corrections, as below indicated.

- line 257 and following "As for M-GLY (Figure 5), the residues at 350 °C clearly showed the spherical cavities due to the volatilization of part of the decomposition products. At 420 °C this phenomenon was more evident, and the dimensions of the cavities increased considerably". However, the morphology are greatly different at 350 and 420 °C and in fig 5 I cannot see increasing of their dimensions. Can you highlight on the figure where are the cavities which size is increasing?

R. This sentence was modified based on the Reviewer's comment.

- lines 353-354 please delete the sentence in bold.

R. The lines were deleted.

- line 408-409. Please cite the reference because the cone calorimeter results are not shown in the present paper.

R. In accordance with the Reviewer’s comment, the reference [7] was cited at line 415.

Reviewer 2 Report

The article studies the science of the chemical transformation happening on cotton fabrics. This type of article may be especially helpful for the cotton industry and to improve environmental sustainability, improving market share for natural fibers, instead of man-made fibers, usually produced from oil. The authors did a good job, but there are small points in the article that must be fixed before approval. The points are detailed below:

*Title: OK

*Abstract: OK

*Keywords: OK

*Introduction:

-The current flow is:

Introduction to the IFRs > PAAs as potential IFRs > PAAs in flame tests > Chemical transformation of PAAs > Use of M-GLY as an IFR and use of instrumental techniques for the study

The flow is not bad, but I would separate the last paragraph in two, and add a paragraph with the objective of this research, changing the flow to:

Introduction to the IFRs > PAAs as potential IFRs > PAAs in flame tests > Chemical transformation of PAAs > Use of M-GLY as an IFR > Instrumental techniques for the study of thermal degradation > Objective of this research

-Line 62: Please, avoid beginning a paragraph with “They”. Do the authors mean the IFRs or the PAAs? Please, replace “They” for the chemical that “…extinguished the flame in horizontal flame spread tests…”.

-Please, clarify if the add-on range is from 4% to 19% w/w or w/v, or something else.

*Material and Methods:

-Please, check if the unit for the cotton fabric density is 200 gm-2. I believe this unit is not corresponding to g/m2. Furthermore, IF possible, explain if the cotton fabric is knitted or woven and if the fabric was scoured.

-Please, add the unit for the molecular weight cutoff of the membrane. Would it be 5000 Da?

-I calculated the yield as 72%: 19.5/(15.4 + 7.5 + 4.2). If I miscalculated the yield, please clarify in the text how the yield was calculated.

-The impregnation strategy is fine, but I would like to suggest, in future research, to try impregnating the cotton fabric with a pad-dry-cure system.

 *Results and Discussions:

-The FT-IR discussion is good, but did not the authors detect any volatile compound with nitrogen? If the authors did or did not, I think it is worth discussing the presence or absence of N-compounds on the volatile phase.

-Lines 353-355: Please, correct the citation. It seems that a problem happened with the bibliographic citation software.

-The authors greatly explained the effects of thermal decomposition on the treated and untreated fabrics. Nevertheless, the authors could discuss more how the nitrogen impacted the thermal decomposition. Please, add this type of discussion in the text too.

*Conclusions: OK

*Supplementary material: OK

*References:

-Please, avoid writing the reference title with all the words in uppercase, even if the document was published like this. For example, write the reference “Polyamidoamines Derived from Natural α-Amino Acids as Effective Flame Retardants for Cotton” as “Polyamidoamines derived from natural α-amino acids as effective flame retardants for cotton”, please.

Author Response

Reviewer 2

The article studies the science of the chemical transformation happening on cotton fabrics. This type of article may be especially helpful for the cotton industry and to improve environmental sustainability, improving market share for natural fibers, instead of man-made fibers, usually produced from oil. The authors did a good job, but there are small points in the article that must be fixed before approval. The points are detailed below:

*Title: OK

*Abstract: OK

*Keywords: OK

*Introduction:

-The current flow is:

Introduction to the IFRs > PAAs as potential IFRs > PAAs in flame tests > Chemical transformation of PAAs > Use of M-GLY as an IFR and use of instrumental techniques for the study.

The flow is not bad, but I would separate the last paragraph in two, and add a paragraph with the objective of this research, changing the flow to:

Introduction to the IFRs > PAAs as potential IFRs > PAAs in flame tests > Chemical transformation of PAAs > Use of M-GLY as an IFR > Instrumental techniques for the study of thermal degradation > Objective of this research

R. In accordance with the Reviewer’s comment, this paragraph was revised, and the objective of this research highlighted.

- Line 62: Please, avoid beginning a paragraph with “They”. Do the authors mean the IFRs or the PAAs? Please, replace “They” for the chemical that “…extinguished the flame in horizontal flame spread tests…”.

R. In accordance with the Reviewer’s comment, “They” was replaced with “PAAs”.

- Please, clarify if the add-on range is from 4% to 19% w/w or w/v, or something else.

R. In accordance with the Reviewer’s comment, this sentence was rewritten for better explaining.

*Material and Methods:

- Please, check if the unit for the cotton fabric density is 200 gm-2. I believe this unit is not corresponding to g/m2. Furthermore, IF possible, explain if the cotton fabric is knitted or woven and if the fabric was scoured.

R. In accordance with the Reviewer’s comment, the characteristics of the cotton fabrics were added and the density unit rewritten.

- Please, add the unit for the molecular weight cutoff of the membrane. Would it be 5000 Da?

R. Yes, the molecular cutoff was 5000 Da. Following the Reviewer’s comment, this info was added.

- I calculated the yield as 72%: 19.5/(15.4 + 7.5 + 4.2). If I miscalculated the yield, please clarify in the text how the yield was calculated.

R. 85% is the correct yield. In fact, the amount of LiOH.H2O was not included in the calculation because it was lost during the purification and isolation procedures. Following the Reviewer’s comment, this info was added in the Experimental part.

-The impregnation strategy is fine, but I would like to suggest, in future research, to try impregnating the cotton fabric with a pad-dry-cure system.

R. Thanks for the suggestion! 

 *Results and Discussions:

-The FT-IR discussion is good, but did not the authors detect any volatile compound with nitrogen? If the authors did or did not, I think it is worth discussing the presence or absence of N-compounds on the volatile phase.

R. In accordance with the Reviewer’s comment, a brief sentence was added to the text at lines 231-233.

- Lines 353-355: Please, correct the citation. It seems that a problem happened with the bibliographic citation software.

R. The sentence was deleted. Thanks for the suggestion!

- The authors greatly explained the effects of thermal decomposition on the treated and untreated fabrics. Nevertheless, the authors could discuss more how the nitrogen impacted the thermal decomposition. Please, add this type of discussion in the text too.

R. In accordance with the Reviewer’s comment, a brief sentence was added to the text at lines 236-237.

*Conclusions: OK

*Supplementary material: OK

*References:

- Please, avoid writing the reference title with all the words in uppercase, even if the document was published like this. For example, write the reference “Polyamidoamines Derived from Natural α-Amino Acids as Effective Flame Retardants for Cotton” as “Polyamidoamines derived from natural α-amino acids as effective flame retardants for cotton”, please.

R. Following the Reviewer’s comment, the title was corrected.